# INTEGRATING KERNEL METHODS AND DEEP NEURAL NETWORKS FOR SOLVING PDES

## ABSTRACT

Physics-informed machine learning (PIML) has emerged as a promising alternative to conventional numerical methods for solving partial differential equations (PDEs). PIML models are increasingly built via deep neural networks (NNs) whose performance is very sensitive to the NN's architecture, training settings, and loss function. Motivated by this limitation, we introduce kernel-weighted Corrective Residuals (CoRes) to integrate the strengths of kernel methods and deep NNs for solving nonlinear PDE systems. To achieve this integration, we design a modular and robust framework which consistently outperforms competing methods in a broad range of benchmark problems. This performance improvement has a theoretical justification and is particularly attractive since we simplify the training process while negligibly increasing the inference costs. Our studies also indicate that the proposed approach considerably decreases the sensitivity of NNs to factors such as random initialization, architecture type, and choice of optimizer.

## 1 INTRODUCTION

Partial differential equations (PDEs) elegantly explain the behavior of many engineered and natural systems such as power grids (Brunton et al., 2016), advanced materials (Mozaffar et al., 2019), and biological agents (Brunton et al., 2016). Since most PDEs cannot be analytically solved, numerical approaches such as the finite element method are frequently used to solve them. Recently, a new class of methods known as physics-informed machine learning (PIML) has been developed and successfully used in many applications.While PIML models have fueled a renaissance in modeling complex systems, their performance heavily depends on optimizing the model's training mechanism, loss function, and architecture (Pestourie et al., 2023). To reduce the time and energy footprint of developing PIML models while improving their accuracy, we re-envision solving PDEs via machine learning (ML) and introduce Corrective Residuals (CoRes) that integrate the strengths of kernel methods and deep neural networks (NNs).

## 2 RELATED WORKS

We can broadly classify PIML models into two categories. The first group of methods relies on variants of neural networks (NNs) and can be traced back to Lagaris et al. (1998). Physics-informed neural networks (PINNs) (Raissi et al., 2019; Sirignano & Spiliopoulos, 2018) and their extensions are the most widely adopted member of these methods and their basic idea is to parameterize the PDE solution via a deep NN. The parameters of this NN are optimized by minimizing a multi-component loss function which encourages the NN to satisfy the PDE as well as the initial and/or boundary conditions (IC and BCs), see Figure A2. This minimization is known to be very sensitive to the optimizer, loss function formulation, and NN's architecture. To decrease this sensitivity, recent works have developed adaptive loss functions (Chen et al., 2018) and tailored architectures that improve gradient flows or automatically satisfy BCs (Wang et al., 2021a; Lagaris et al., 1998). These advancements, however, fail to generalize to a diverse set of PDEs and substantially increase the cost and complexity of training.

The second group of PIML models leverage kernel methods which have long been used in ML but their application in solving PDEs is largely unexplored. The few existing works (see Zhang et al. (2022); Iwata & Ghahramani (2017); Meng & Yang (2023)) exclusively employ zero-mean Gaussian

processes (GPs). With this choice, solving the PDE amounts to designing the GP's kernel whose parameters are obtained via either maximum likelihood estimation (MLE) or a regularized MLE where the penalty term quantifies the GP's error in satisfying the PDE system. In a recent work by Chen et al. (2021), solving PDEs via a zero-mean GP is cast as an optimal recovery problem that aims to estimate the solution at a finite number of interior nodes in the domain. Once these values are estimated, the PDE solution is approximated anywhere in the domain via kernel regression.

## 3   Neural Networks with Corrective Residuals

Kernel methods such as GPs have less extrapolation and scalability powers compared to deep NNs. They also struggle to approximate PDE solutions that have large gradients or involve coupled dependent variables. However, GPs locally generalize better than NNs and are interpretable and easy to train. Grounded on these properties, we introduce deep architectures with kernel-weighted CoRes that integrate the attractive features of NNs and GPs for solving PDEs.

### 3.1   Theoretical Rationale of the Proposed Approach

We argue that the sole reliance on the kernel serves as a double-edged sword when solving PDEs. To demonstrate, we consider the task of emulating the function $u(\mathbf{x})$ given the $n$ samples $\mathbf{X} = \{\mathbf{x}_1, \cdots, \mathbf{x}_n\}$ with corresponding outputs $\mathbf{u} = \{u_1, \cdots, u_n\}$ where $u_i = u(\mathbf{x}_i)$. If we endow $u(\mathbf{x})$ with a GP prior with the mean function $m(\mathbf{x}; \boldsymbol{\theta})$ and kernel $c(\mathbf{x}, \mathbf{x}'; \boldsymbol{\phi})$, the conditional process is also a GP whose expected value at $\mathbf{x}^*$ is:

$$\eta(\mathbf{x}^*; \boldsymbol{\theta}, \boldsymbol{\phi}) := \mathbb{E}[u^* | \mathbf{u}, \mathbf{X}] = m(\mathbf{x}^*; \boldsymbol{\theta}) + \boldsymbol{w}^T \boldsymbol{r}, \tag{1a}$$

$$\boldsymbol{w} := w(\mathbf{x}^*, \mathbf{X}; \boldsymbol{\phi}) = c^{-1}(\mathbf{X}, \mathbf{X}; \boldsymbol{\phi}) c(\mathbf{X}, \mathbf{x}^*; \boldsymbol{\phi}) \tag{1b}$$

$$\boldsymbol{r} := r(\mathbf{X}, \mathbf{u}; \boldsymbol{\theta}) = \mathbf{u} - m(\mathbf{X}; \boldsymbol{\theta}). \tag{1c}$$

Here, $\boldsymbol{\theta}$ and $\boldsymbol{\phi}$ are the model parameters (typically estimated via MLE), $c(\mathbf{X}, \mathbf{x}^*; \boldsymbol{\phi}) = [c(\mathbf{x}_1, \mathbf{x}^*; \boldsymbol{\phi}), \cdots, c(\mathbf{x}_n, \mathbf{x}^*; \boldsymbol{\phi})]^T$, $\boldsymbol{r}$ denotes the *residuals* on the training data, $\boldsymbol{w}$ are the kernel-induced weights, and $\boldsymbol{C} = c(\mathbf{X}, \mathbf{X}; \boldsymbol{\phi})$ is the covariance matrix with $ij^{th}$ entry $c(\mathbf{x}_i, \mathbf{x}_j; \boldsymbol{\phi})$. The covariance function can be a deep NN as in Wilson et al. (2016) or the simple Gaussian kernel:

$$c(\mathbf{x}, \mathbf{x}'; \boldsymbol{\phi}) = \exp\big\{-(\mathbf{x} - \mathbf{x}')^T \operatorname{diag}(\boldsymbol{\phi})(\mathbf{x} - \mathbf{x}')\big\}. \tag{2}$$

Since many kernels can approximate an arbitrary continuous function (Rasmussen, 2006), zero-mean GPs are used in many regression problems as eliminating $m(\mathbf{x}; \boldsymbol{\theta})$ reduces the number of trainable parameters while increasing numerical stability (see Appendix A 1 for more detailed discussions). Unlike regression, PDE systems cannot be accurately solved via zero-mean GPs without any in-domain samples since the posterior process in Equation (1) predicts zero for any point that is sufficiently far from the boundaries. This reversion to the mean behavior is due to the exponential decay of the correlations as the distance between two points increases, see Equation (2).

Following the above discussions, we make two important observations on Equation (1): it heavily relies on $m(\mathbf{x}; \boldsymbol{\theta})$ in data scarce regions and it regresses $\mathbf{u}$ regardless of the values of $m(\mathbf{X}; \boldsymbol{\theta})$. These observations suggest that a GP with an NN mean function provides an attractive *prior* for solving PDEs since functions formulated as in Equation (1a) satisfy the BCs/IC and their smoothness can be controlled through the mean and covariance functions. This approach, however, presents two major challenges. First, the posterior distribution in this case will *not* be Gaussian in general since most practical PDEs are nonlinear. Second, jointly optimizing $\boldsymbol{\phi}$ and $\boldsymbol{\theta}$ is a computationally expensive and unstable process due to the repeated need for constructing and inverting $\boldsymbol{C}$.

### 3.2   Proposed Framework

We address the above challenges via modularization and formulating the training process based on maximum a posteriori (MAP) instead of MLE. Our framework consists of two sequential modules that aim to solve PDE systems with deep NNs that substantially benefit from kernel-weighted CoRes. These modules seamlessly integrate the best of two worlds: (1) the local generalization power of kernels close to the domain boundaries, and (2) the substantial capacity of deep NNs in learning multiple levels of distributed representations in the interior regions where there are no labeled data.

In the first module, we endow the PDE solution with a GP prior whose mean and covariance functions are a deep NN and the Gaussian kernel in Equation (2), respectively. Conditioned on the data, $\mathbf{u}$, the posterior distribution of the solution is again a GP and follows Equation (1a) where $r$ and $w$ denote the residuals and kernel-induced weights, respectively. Importantly, in this module we fix $\boldsymbol{\theta}$ to some random values and choose $\widehat{\boldsymbol{\phi}}$ such that the GP can faithfully reproduce $\mathbf{u}$.

In the second module, we obtain the final model by conditioning the GP on the (nonlinear) constraints that require the in-domain predictions to satisfy the PDE system. We achieve this conditioning by fixing $\widehat{\boldsymbol{\phi}}$ from module 1 and optimizing $\boldsymbol{\theta}$ to ensure that the model in Equation (1a) satisfies the PDE at $n_{PDE}$ randomly selected collocation points (CPs) in the domain, see Figure 1.

**Model Characteristics:** As extensively studied in Appendix A, our approach provides four unique features. First, the training cost in our approach mainly depends on the second module since selecting $\widehat{\boldsymbol{\phi}}$ does not rely on MLE and is an inexpensive process. Additionally, our experiments consistently indicate that the performance of the final model is quite robust to $\widehat{\boldsymbol{\phi}}$ as long as the BCs/IC are sufficiently sampled. This robustness is independent of the random values assigned to $\boldsymbol{\theta}$ in module 1. Based on these two observations, in all of our experiments we simply assign $10^2$ to all the kernel parameters and sample $40$ points at each boundary.

Second, the computational cost of coupling GPs and deep NNs in our framework is negligible during both training and testing since $C$ does not change in the second module and its size only depends on $\mathbf{u}$. When solving PDEs such as the Navier-Stokes equations that have multiple dependent variables, the size of $\mathbf{u}$ can grow rapidly since it will store boundary and initial data on multiple outputs. Hence, for such PDEs we decouple the kernel-weighted CoRes of the outputs to keep $C$ and $\mathbf{u}$ small. As shown in Figure A5, we formulate this decoupling by endowing the dependent variables with a collection of GP priors which share the same mean function but have independent kernels.

The third feature of our model is its ability to exactly satisfy the BCs/IC as the number of sampled boundary points increases. This behavior is independent of the domain geometry and the potential noise corrupting the data; see Appendix A for the proof and examples. Due to this feature, the loss function in Figure 1 only minimizes the error in satisfying the PDE and excludes data loss terms.

The above three features imply that training a PINN or its extensions costs similarly to the case where the same network is used as $m(\mathbf{x}; \boldsymbol{\theta})$ in our framework. This behavior is very attractive since our approach consistently and substantially improves the performance of existing NN-based methods while also simplifying the training process.

Lastly, our framework allows to perform data fusion and system identification with fast convergence rates by incorporating the additional measurements in the kernel structure in exactly the same way that BC/IC data are handled, see Appendix A 5.5 for details and results.

## 4 RESULTS

We compare the performance of our approach against four methods on the PDE systems given by Equations (11) to (14) in Appendix A 3. We study each problem under two settings to understand

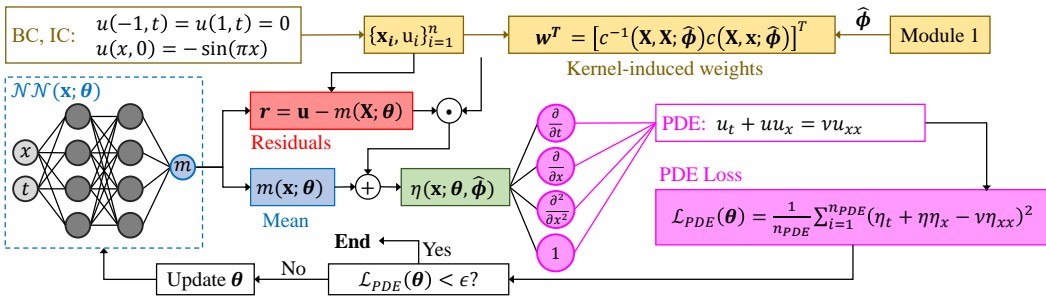

Figure 1: Flowchart of module two: We estimate $\boldsymbol{\theta}$ for the 1D Burgers' equation. The loss function only depends on the PDE since BC/IC are automatically satisfied.

the effect of PDE complexity on the results. We obtain the reference solution $u(\mathbf{x})$ for each PDE system as detailed in Appendix A 4.5 and use it to quantify the accuracy of the PIML models based on the Euclidean error norm, $L_e^2$, at $n_t = 10^4$ randomly chosen points. Throughout, $[x, y]$, $t$, and $u$ denote the spatial coordinates, time, and the PDE solution, respectively. In the case of Navier-Stokes equation, $\mathbf{u}(\mathbf{x}) = [u(\mathbf{x}), v(\mathbf{x}), p(\mathbf{x})]^T$ denotes the solution.

As detailed in Appendix A 4, we use a simple feed-forward NN in our framework and design its input and output dimensionality based on the PDE system. We denote our model via NN-CoRes and compare it against (1) GP$_{\text{OR}}$ of Chen et al. (2021), (2) PINNs whose architectures are exactly the same as the mean function of NN-CoRes, (3) PINN$_{\text{DW}}$ which leverages dynamic weights for loss terms, and (4) PINN$_{\text{HC}}$ which is a PINN whose output is designed to strictly satisfy the BCs/IC.

The results of our studies are summarized in Table 1 and indicate that our approach consistently outperforms other methods by relatively large margins. Interestingly, in most cases even the small NN-CoRes achieve lower errors than the high capacity version of the competing methods; indicating NN-CoRes more effectively use their networks' capacity to learn the PDE solution. To visually compare the capacity utilization across different NN-based models, in Figure A12 we provide the histogram of the PDE loss gradients with respect to $\boldsymbol{\theta}$ at the end of training. We observe that NN-CoRes achieve the most near-zero gradients *while* satisfying the BCs/IC. In contrast, PINN$_{\text{HC}}$, which is also designed to automatically satisfy the BCs/IC, struggles to minimize the PDE loss.

We observe in Table 1 that the performance of all the methods drops as either the problem complexity increases (e.g., Burger's vs. LDC) or PDE parameters are changed to introduce nonlinearity (e.g., $A = 3$ vs $A = 5$ in LDC). This trend is expected as the architecture and training settings across our experiments are fixed. That is, we can increase the accuracy of all methods, especially NN-CoRes, by increasing their capacity or improving the training, see Appendix A 5.3 for multiple experiments.

## 5 CONCLUSIONS AND FUTURE WORKS

We introduce kernel-weighted CoRes that integrate the strengths of kernel methods and deep NNs for solving nonlinear PDEs. We design a modular framework to achieve this integration and show that it improves the accuracy without complicating the training process. As extensively studied in Appendix A, our findings not only are very robust to the choice of optimizer and initial parameter values, but also applicable to various neural architectures.

The current major limitation of our approach is that the contributions of the kernel-weighted CoRes decrease in the absence of boundary data. We believe devising periodic kernels is a promising direction for addressing this limitation which will be particularly useful in multiscale simulations where PDEs with periodic BCs frequently arise in the fine-scale analyses.

ACKNOWLEDGMENTS

We appreciate the support from the Office of the Naval Research, NASA's Space Technology Research Grants Program, and National Science Foundation.

Table 1: Summary of comparative studies: We report $L_e^2 \times 10^3$ of different methods as a function of model capacity and PDE parameter. The symbol $\otimes$ indicates the network architecture (e.g., $4 \otimes 10$ is an NN which has four $10-$ neuron hidden layers). Unlike NN-based methods, GP$_{\text{OR}}$'s accuracy relies on the number of interior nodes which we set to $1,000$ or $2,000$.

| Problem | Capacity | NN-CoRes | | GP$_{\text{OR}}$ | | PINN | | PINN$_{\text{DW}}$ | | PINN$_{\text{HC}}$ | |
|---|---|---|---|---|---|---|---|---|---|---|---|
| | | $4 \otimes 10$ | $4 \otimes 20$ | 1,000 | 2,000 | $4 \otimes 10$ | $4 \otimes 20$ | $4 \otimes 10$ | $4 \otimes 20$ | $4 \otimes 10$ | $4 \otimes 20$ |
| Burger's | $\nu = 0.02/\pi$ | **0.80** | 0.89 | 224 | 169 | 2.42 | 1.50 | 2.86 | 3.18 | 341 | 336 |
| | $\nu = 0.01/\pi$ | 1.91 | **1.29** | 169 | 208 | 4.26 | 4.38 | 19.3 | 5.79 | 365 | 352 |
| Elliptic | $\alpha = 20$ | 4.50 | **2.04** | 2.44 | 4.06 | 655 | 468 | 297 | 126 | 635 | 595 |
| | $\alpha = 30$ | 4.38 | **1.24** | 7.08 | 6.55 | 845 | 555 | 169 | 119 | 289 | 653 |
| Eikonal | $\epsilon = 0.05$ | 0.52 | **0.37** | 176 | 125 | 2.76 | 2.19 | 2.01 | 1.54 | 289 | 288 |
| | $\epsilon = 0.01$ | **4.60** | 4.99 | 218 | 206 | 6.41 | 6.38 | 5.03 | 4.97 | 291 | 342 |
| LDC | $A = 3$ | 186 | **86.7** | — | — | 272 | 128 | 301 | 125 | 432 | 529 |
| | $A = 5$ | 311 | **279** | — | — | 717 | 677 | 716 | 623 | 1028 | 911 |

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

## A  APPENDIX

### A 1  PROPERTIES OF A GAUSSIAN PROCESS SURROGATE

We use an analytic one-dimensional ($1D$) function to demonstrate some of the most important characteristics of Gaussian process (GP) surrogates. Specifically, we leverage a set of examples to argue that GPs: (1) have interpretable parameters, (2) can regress or interpolate highly nonlinear functions, (3) suffer from reversion to the mean phenomena in data scarce regions, (4) can have ill-conditioned covariance matrices if their mean function interpolates the data, and (5) with manually chosen hyperparameters can faithfully surrogate a function if sufficient training samples are available. These properties underpin our decision for manually selecting the kernel parameters in module one of our framework. They also demonstrate the effects of a GP's mean function on its prediction power and numerical stability.

As demonstrated in Figure A1 our experiments involve sampling from a sinusoidal function where we study the effects of frequency, noise, data distribution, function differentiability, adopted prior mean function, and hyperparameter optimization on the behavior of GPs. For all of these studies we endow the GP with the following parametric kernel:

$$c(x, x'; \phi, \delta, \sigma^2) = \sigma^2 \exp\{-\phi(x - x')^2\} + \mathbb{1}\{x == x'\}\delta, \tag{3}$$

where $\boldsymbol{\lambda} = [\sigma, \phi, \delta]^T$ are the kernel parameters. In this equation, $\sigma^2$ is the process variance which, looking at Equation 1 in the main text, does not affect the posterior mean and hence we simply set it to $1$ in our framework (this feature of our framework is in sharp contrast to other methods such as Chen et al. (2021) whose performance is quite sensitive to the selected kernel parameters). The

rest of the parameters in Equation (3) are defined as follows. $\phi = 10^\omega$ where $\omega$ is the length-scale or roughness parameter that controls the correlation strength along the $x-$axis, $\mathbb{1}\{\cdot\}$ returns $1/0$ if the enclosed statement is true/false, and $\delta$ is the so-called nugget or jitter parameter that is added to the kernel for modeling noise and/or improving the numerical stability of the covariance matrix. We quantify the numerical stability of the covariance matrix via its condition number or $\kappa$.

Given some training data, $\boldsymbol{\lambda}$ can be quickly estimated via maximum likelihood estimation (MLE). We denote parameter estimates obtained via this process by appending the subscript MLE to them, i.e., $\widehat{\boldsymbol{\lambda}}_{MLE}$. Alternatively, we can manually assign specific values to $\boldsymbol{\lambda}$.

We first study the effect noise by training two GPs where both GPs aim to emulate the same underlying function but one has access to noise-free responses while the other is trained on noisy data, see Figure A1 (a) and Figure A1 (b), respectively. We observe in Figure A1 (a) that the estimated value for $\widehat{\delta}_{MLE}$ is very small since the data is noise-free (the small value is added to reduce $\kappa$) while in Figure A1 (b) the estimated nugget parameter is much larger and close to the noise variance ($2.48e - 3$ vs. $2.50e - 3$). Additionally, comparing Figure A1 (a) and Figure A1 (c) we observe a direct relation between the frequency of the underlying function and the estimated kernel parameters. In particular, the magnitude of $\widehat{\omega}_{MLE}$ increases as $u(x)$ becomes rougher since the correlation between two points on it quickly dies out as the distance between those points increases (for this reason, $\omega$ is also sometimes called the roughness parameter). Further increasing the frequency of $u(x)$ to the extend that it resembles a noise signal directly increases $\widehat{\omega}_{MLE}$. These points indicate that the kernel parameters of a GP are interpretable.

We next study the reversion to the mean behavior and numerical instabilities of GPs in Figure A1 (d) and Figure A1 (e). In both of these scenarios the training data is only available close to the boundaries. However, we set the prior mean of the GP in Figure A1 (d) and Figure A1 (e) to zero and $m(x; \theta) = \theta \times \sin(2\pi x)$, respectively. The reversion to the mean behavior is clearly observed in Figure A1 (d) where the expected value of the posterior distribution is almost zero in the $(-0.5, 0.5)$ range where the correlations with the training data die out. The reversion to the mean behavior is also seen in Figure A1 (e) but this time it is not undesirable since the functional form of the chosen parametric mean function is similar to $u(x)$ (note that a large neural network can also reproduce the training data but such a network cannot match $u(x)$ in interior regions where there are no labeled data). This similarity forces the kernel to regress residuals that are mostly zero (i.e., the kernel must regress a constant value in the entire domain). Since any two points on a constant function have maximum correlation, regressing such residuals requires $\phi \to 0$ which, in turn, renders the covariance matrix ill-conditioned to the extend that $\kappa \to +\infty$. Based on these observations, in our framework we do not estimate the kernel parameters jointly with the weights and biases of the deep neural network (NN).

Lastly, in Figure A1 (e) we demonstrate that GPs can interpolate non-differentiable functions as long as they are provided with sufficient training data. The power and efficiency of GPs in learning from data is quite robust to the hyperparameters. As shown in Figure A1 (g) through Figure A1 (i) GPs with manually selected $\omega$ can accurately surrogate $u(x)$ regardless of its frequency (the nugget value in these three cases is chosen such that $\kappa$ does not exceed a pre-determined value). This attractive behavior forms the basis of our choice to manually fix $\phi$ in the first module of our framework. It is highlighted that the manual parameter selection results in sub-optimal prediction intervals but this issue does not affect our framework since we do not leverage these intervals.

## A 2    NEURAL NETWORKS WITH KERNEL-WEIGHTED CORRECTIVE RESIDUALS REPRODUCE THE DATA

We prove that the error of our model in reproducing the boundary data converges to zero as we increase the number of sampled boundary data. For the sake of completeness, we begin by a definition and invoking two theorems and then proceed with our proof.

**Reproducing kernel Hilbert space (RKHS):** Let $\mathcal{H}$ be a Hilbert space of real functions u defined on an index set $\mathcal{X}$. Then, $\mathcal{H}$ is called an RKHS with the inner product $\langle \cdot, \cdot \rangle_{\mathcal{H}}$ if the function $c : \mathcal{X} \times \mathcal{X} \to \mathbb{R}$ with the following properties exists:

- For any $\mathbf{x}$, $c(\mathbf{x}, \mathbf{x}')$ as a function of $\mathbf{x}'$ is in $\mathcal{H}$,
- $c$ has the reproducing property, that is $\langle u(\mathbf{x}'), c(\mathbf{x}', \mathbf{x}) \rangle_{\mathcal{H}} = u(\mathbf{x})$.

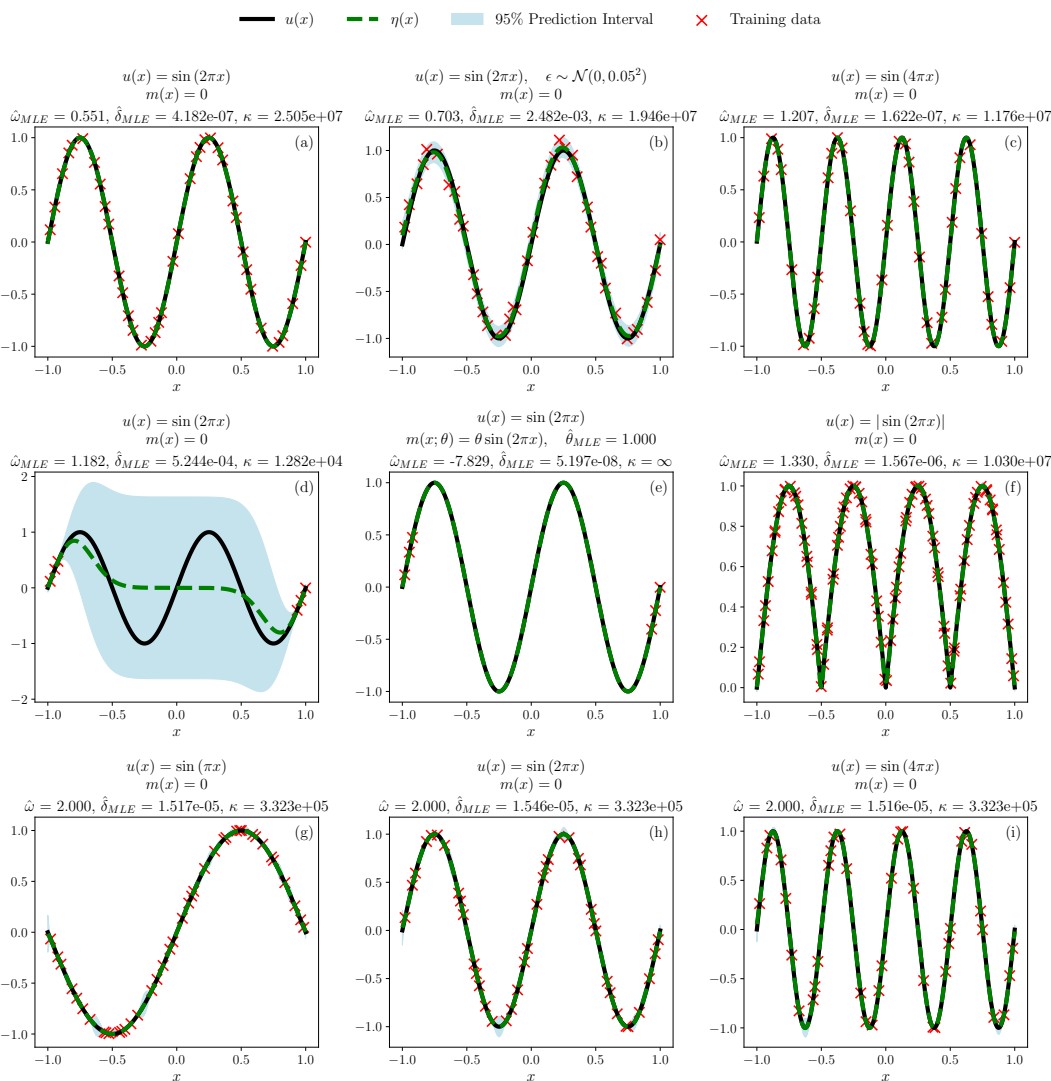

Figure A1: Properties of Gaussian processes: We demonstrate that GPs have interpretable hyperparameters and can regress a wide range of functions. The dependency of regression quality on the hyperparameters rapidly decreases as the size of the training data increases.

Note that the norm of $u$ is $\| u \|_{\mathcal{H}} = \sqrt{\langle u, u \rangle_{\mathcal{H}}}$ and that $\langle c(\mathbf{x}, \cdot), c(\mathbf{x}', \cdot) \rangle_{\mathcal{H}} = c(\mathbf{x}, \mathbf{x}')$ since both $c(\mathbf{x}', \cdot)$ and $c(\mathbf{x}, \cdot)$ are in $\mathcal{H}$.

**Mercer's Theorem:** The eigenfunctions of the real positive semidefinite kernel $c(\mathbf{x}, \mathbf{x}')$ whose eignenfunction expansion with respect to measure $\pi$ is $c(\mathbf{x}, \mathbf{x}') = \sum_{i=1}^{N} \alpha_i \psi_i(\mathbf{x}) \psi_i(\mathbf{x}')$, are orthonormal. That is:

$$\int \psi_i(\mathbf{x}) \psi_j(\mathbf{x}) d\pi = \delta_{ij} \tag{4}$$

where $\delta_{ij}$ denotes the Kronecker delta function. Following this theorem, we note that for a Hilbert space defined by the linear combinations of the eigenfunctions, that is $u(\mathbf{x}) = \sum_{i=1}^{N} u_i \psi_i(\mathbf{x})$ with $\sum_{i=1}^{N} u_i / \alpha_i < \infty$, we have $\| u \|_{\mathcal{H}}^2 = \langle u, u \rangle_{\mathcal{H}} = \sum_{i=1}^{N} u_i / \alpha_i$.

**Representer Theorem:** Each minimizer $u(\mathbf{x}) \in \mathcal{H}$ of the following functional can be represented as $u(\mathbf{x}) = \sum_{i=1}^{n} \alpha_i c(\mathbf{x}, \mathbf{x}_i)$:

$$F[u(\mathbf{x})] = \frac{\beta}{2} \| u(\mathbf{x}) \|_{\mathcal{H}}^2 + P(\mathbf{h}, \mathbf{u}). \tag{5}$$

where $\boldsymbol{h} = [h_1, \cdots, h_n]^T$ is the observation vector, $u(\mathbf{x})$ denotes the function that we aim to fit to $\boldsymbol{h}$, $\mathbf{u} = [u(\mathbf{x}_1), \cdots, u(\mathbf{x}_n)]^T = [u_1, \cdots, u_n]^T$ are the evaluations of $u(\mathbf{x})$ at configurations where $\boldsymbol{h}$ are observed, $\beta$ is a scaling constant that balances the contributions of the two terms on the right hand side (RHS) to $F[u(\mathbf{x})]$, and $P(\cdot, \cdot)$ is a function that evaluates the quality of $u(\mathbf{x})$ in reproducing $\boldsymbol{h}$. For proof of this theorem, see Schölkopf & Smola (2002); O'sullivan et al. (1986); Kimeldorf & Wahba (1971).

In our case, to prove that our model can reproduce the boundary data we first assume that the initial and boundary conditions are sufficiently smooth functions and that the neural network (i.e., the mean function of the GP) produces finite values on the boundaries. These assumptions simplify the proof by allowing us to work with the difference of these two terms.

We now consider a specific form of Equation (5):

$$F[u(\mathbf{x})] = \frac{1}{2} \parallel u(\mathbf{x}) \parallel_{\mathcal{H}}^2 + \frac{\lambda^2}{2} \sum_{i=1}^{n} (h_i - u(\mathbf{x}_i))^2, \tag{6}$$

where $u(\mathbf{x})$ is the zero-mean GP predictor and $\parallel u(\mathbf{x}) \parallel_{\mathcal{H}}$ is the RKHS norm with kernel $c(\cdot, \cdot)$. The second term on the right hand side corresponds to the negative log-likelihood of a Gaussian noise model with precision $\lambda^2$ and hence the minimizer of Equation (6) is the posterior mean of the GP Szeliski (1987). Hence, we now need to show that as $n \to \infty$ the minimizer of Equation (6), which is our GP, can reproduce the data $\boldsymbol{h}$. We denote the ground truth function that we aim to discover and the variance around it by, respectively, $h(\mathbf{x})$ and $\tau^2(\mathbf{x}) = \int (h - h(\mathbf{x}))^2 d\pi(h|\mathbf{x})$ where $\pi(\mathbf{x}, h)$ is the probability measure that generates the data $(\mathbf{x}_i, h_i)$.

We rewrite the second term on the right hand side of Equation (6) as:

$$\mathbb{E}\left[\sum_{i=1}^{n} (h_i - u(\mathbf{x}_i))^2\right] = n \int (h - u(\mathbf{x}))^2 d\pi(\mathbf{x}, h) =$$

$$n \int (h - h(\mathbf{x}) + h(\mathbf{x}) - u(\mathbf{x}))^2 d\pi(\mathbf{x}, h) = \tag{7}$$

$$n \int \tau^2(\mathbf{x}) d\pi(\mathbf{x}) + 0 + n \int (h(\mathbf{x}) - u(\mathbf{x}))^2 d\pi(\mathbf{x}).$$

where the zero on the last line is due to the definition of $h(\mathbf{x})$, i.e., $h(\mathbf{x}) = \mathbb{E}[h|\mathbf{x}]$. Since $\tau^2(\mathbf{x})$ is independent of $u(\mathbf{x})$, we can use Equation (7) to rewrite Equation (6) as:

$$F_\pi[u(\mathbf{x})] = \frac{1}{2} \parallel u(\mathbf{x}) \parallel_{\mathcal{H}}^2 + \frac{n\lambda^2}{2} \int (h(\mathbf{x}) - u(\mathbf{x}))^2 d\pi(\mathbf{x}). \tag{8}$$

We now invoke Mercer's theorem to write $u(\mathbf{x}) = \sum_{i=1}^{\infty} u_i \psi_i(\mathbf{x})$ and $h(\mathbf{x}) = \sum_{i=1}^{\infty} h_i \psi_i(\mathbf{x})$ where $\psi_i$ are the eigenfunctions of the nondegenerage kernel of the GP. Since $\{\psi_i\}$ form an orthonormal basis, we can write:

$$F_\pi[u(\mathbf{x})] = \frac{1}{2} \sum_{i=1}^{\infty} \frac{u_i^2}{\alpha_i} + \frac{n\lambda^2}{2} \sum_{i=1}^{\infty} (h_i - u_i)^2. \tag{9}$$

We take the derivative of Equation (9) with respect to $u_i$ and set it to zero to obtain:

$$u_i = \frac{\alpha_i h_i}{\alpha_i + 1/n\lambda^2}. \tag{10}$$

Since $1/n\lambda^2 \to 0$ as $n \to \infty$, in the limit $u_i \to h_i$, i.e., our zero-mean GP predictor corrects for the error that $m(\mathbf{x}, \boldsymbol{\theta})$ has on reproducing the initial and boundary conditions. Note that the convergence in Equation (10) does not depend on $\tau^2(\mathbf{x})$ and hence holds for the case where the observation vector $\boldsymbol{h}$ is noisy.

## A 3   DETAILS ON THE BENCHMARK PROBLEMS

Below, we provide the governing equations as well as the initial and boundary conditions for the four PDE systems studied in this paper.

**Burgers' Equation** We consider a viscous system subject to IC and Dirichlet BC in one space dimension:

$$
\begin{aligned}
u_t + uu_x - \nu u_{xx} &= 0, && \forall x \in (-1,1), t \in (0,1] \\
u(-1,t) = u(1,t) &= 0, && \forall t \in [0,1] \\
u(x,0) &= -\sin(\pi x), && \forall x \in [-1,1]
\end{aligned}
\tag{11}
$$

where $\mathbf{x} = [x,t]$ and $\nu$ is the kinematic viscosity. Equation (11) frequently arises in fluid mechanics and nonlinear acoustics. In our studies, we investigate the performance of different PIML models in solving Equation (11) for $\nu = \left\{ \frac{0.01}{\pi}, \frac{0.02}{\pi} \right\}$ which controls the solution smoothness at $x = 0$ where a shock wave forms as $\nu$ approches zero.

**Nonlinear Elliptic PDE** To assess the ability of our approach in learning high-frequency solutions, we study the boundary value problem developed in Chen et al. (2021):

$$
\begin{aligned}
u_{xx} + u_{yy} - \alpha u^3 &= f(x,y), && \forall x, y \in (0,1)^2 \\
u(x,0) = u(x,1) &= 0, && \forall x \in [0,1] \\
u(0,y) = u(1,y) &= 0, && \forall y \in [0,1]
\end{aligned}
\tag{12}
$$

where $\mathbf{x} = [x,y]$ and $\alpha = \{20, 30\}$ is a constant that controls the nonlinearity degree. $f(x,y)$ is designed such that the solution is $u(x,y) = \sin(\pi x)\sin(\pi y) + 2\sin(4\pi x)\sin(4\pi y)$.

**Eikonal Equation** We consider the two-dimensional regularized Eikonal equation Chen et al. (2021) which is typically encountered in the context of wave propagation:

$$
\begin{aligned}
u_x^2 + u_y^2 - \epsilon(u_{xx} + u_{yy}) &= 1, && \forall x, y \in (0,1)^2 \\
u(x,0) = u(x,1) &= 0, && \forall x \in [0,1] \\
u(0,y) = u(1,y) &= 0, && \forall y \in [0,1]
\end{aligned}
\tag{13}
$$

where $\mathbf{x} = [x,y]$ and $\epsilon = \{0.01, 0.05\}$ is a constant that controls the smoothing effect of the regularization term.

**Lid-Driven Cavity (LDC)** The two-dimensional steady state LDC problem has become a gold standard for evaluating the ability of PIML models in solving coupled PDEs. This problem is governed by the incompressible Navier-Stokes equations:

$$
\begin{aligned}
u_x + v_y &= 0, && \forall \mathbf{x} \in (0,1)^2 \\
uu_x + vu_y &= -\frac{1}{\rho}p_x + \nu(u_{xx} + u_{yy}), && \forall \mathbf{x} \in (0,1)^2 \\
uv_x + vv_y &= -\frac{1}{\rho}p_y + \nu(v_{xx} + v_{yy}), && \forall \mathbf{x} \in (0,1)^2 \\
v(x,0) = v(x,1) = v(0,y) = v(1,y) &= 0, && \forall x, y \in [0,1] \\
u(x,0) = u(0,y) = u(1,y) &= 0, && \forall x, y \in [0,1] \\
u(x,1) &= A\sin(\pi x), && \forall x \in [0,1] \\
p(0,0) &= 0
\end{aligned}
\tag{14}
$$

where $\mathbf{x} = [x,y]$, $\nu = 0.01$ is the kinematic viscosity, $\rho = 1.0$ denotes the density, and $A = \{3, 5\}$ is a scaling constant. The Reynolds number for this LDC problem can be computed via $Re = \frac{\rho \bar{u} L}{\nu}$ where $\bar{u} = \int_0^1 A\sin(\pi x)dx$ is the characteristic speed of the flow and $L = 1$ is the characteristic length. For the two cases $A = \{3, 5\}$, we obtain $Re = \{191, 318\}$.

A 4  METHODS AND IMPLEMENTATION DETAILS

We first briefly introduce the four PIML models that we have used in our comparative studies and then provide some details on how the reference solution for each PDE system is obtained. These solutions are used to quantify the accuracy of the PIML models.

To be able to directly compare the implementation of the four PIML models, we use Burgers' equation in the following descriptions. The PDE system is:

$$u_t + uu_x - \nu u_{xx} = 0, \qquad \forall x \in [-1, 1], t \in (0, 1] \qquad (15a)$$
$$u(-1, t) = u(1, t) = 0, \qquad \forall t \in [0, 1] \qquad (15b)$$
$$u(x, 0) = -\sin(\pi x), \qquad \forall x \in [-1, 1] \qquad (15c)$$

where $\mathbf{x} = [x, t]$ are the independent variables, $u$ is the PDE solution, and $\nu$ is a constant that denotes the kinematic viscosity. Also, we denote the output of the NN models via $m(\mathbf{x}; \boldsymbol{\theta})$ throughout this section. Note that we also employ $m(\mathbf{x}; \boldsymbol{\theta})$ for denoting the NN in the mean function of NN-CoRes.

### A 4.1   Physics-informed Neural Networks (PINNs)

As schematically shown in Figure A2, the essential idea of PINNs is to parameterize the relation between $u$ and $\mathbf{x}$ with a deep NN Raissi et al. (2019), i.e., $u(\mathbf{x}) = m(\mathbf{x}; \boldsymbol{\theta})$ where $\boldsymbol{\theta}$ are the network's weights and biases. The parameters of $m$ are optimized by iteratively minimizing a loss function, denoted by $\mathcal{L}(\boldsymbol{\theta})$, that encourages the network to satisfy the PDE system in Equation (15). To calculate $\mathcal{L}(\boldsymbol{\theta})$, we first obtain the network's output at $n_{BC}$ points on the $x = -1$ and $x = 1$ boundaries, $n_{IC}$ points on the $t = 0$ boundary which marks the initial condition, and $n_{PDE}$ collocation points (CPs) inside the domain, see Figure A2b. For the $n_{BC} + n_{IC}$ points on the boundaries, we can directly compare the network's outputs to the specified boundary and initial conditions in Equations (15b) and (15c). For each of the $n_{PDE}$ CPs, we evaluate the partial derivatives of the output and calculate the residual in Equation (15a). Once these three terms are calculated, we obtain $\mathcal{L}(\boldsymbol{\theta})$ by summing them up as follows:

$$
\begin{aligned}
\mathcal{L}(\boldsymbol{\theta}) = {} & \mathcal{L}_{PDE}(\boldsymbol{\theta}) + \mathcal{L}_{BC}(\boldsymbol{\theta}) + \mathcal{L}_{IC}(\boldsymbol{\theta}) \\
= {} & \frac{1}{n_{PDE}} \sum_{i=1}^{n_{PDE}} (m_t(\mathbf{x}_i; \boldsymbol{\theta}) + m(\mathbf{x}_i; \boldsymbol{\theta}) m_x(\mathbf{x}_i; \boldsymbol{\theta}) - \nu m_{xx}(\mathbf{x}_i; \boldsymbol{\theta}))^2 + \\
& \frac{1}{n_{BC}} \sum_{i=1}^{n_{BC}} (m(\mathbf{x}_i; \boldsymbol{\theta}) - 0)^2 + \frac{1}{n_{IC}} \sum_{i=1}^{n_{IC}} (m(\mathbf{x}_i; \boldsymbol{\theta}) + \sin(\pi x_i))^2
\end{aligned}
\qquad (16)
$$

The loss function in Equation (16) is typically minimized via either the Adam Kingma & Ba (2014) or L-BFGS Liu & Nocedal (1989) methods which are both gradient-based optimization algorithms. With either Adam or L-BFGS, the parameters of the network are first initialized and then iteratively updated to minimize $\mathcal{L}(\boldsymbol{\theta})$. These updates rely on partial derivaties of $\mathcal{L}(\boldsymbol{\theta})$ with respect to $\boldsymbol{\theta}$ which can be efficiently obtained via automatic differentiation Baydin et al. (2018).

While Adam and L-BFGS are both gradient-based optimization techniques, they have some major differences Sun et al. (2019). Adam is a first-order method while L-BFGS is not since it is a quasi-Newton optimization algorithm. Compared to Adam, L-BFGS is more memory-intensive and has a higher per-epoch computational cost since it uses an approximation of the Hessian matrix during the optimization. Moreover, Adam scales to large datasets better than L-BFGS which does not accommodate mini-batch training. However, L-BFGS typically provides lower loss values and requires fewer number of epochs for convergence compared to Adam.

### A 4.2   Physics-informed Neural Networks With Dynamic Loss Weights

One of the challenges associated with minimizing the loss function in Equation (16) is that the three terms on the right-hand side disproportionately contribute to $\mathcal{L}(\boldsymbol{\theta})$. To mitigate this issue, a popular approach is to scale each loss component independently before summing them up, that is:

$$\mathcal{L}(\boldsymbol{\theta}) = \mathcal{L}_{PDE}(\boldsymbol{\theta}) + w_{BC}\mathcal{L}_{BC}(\boldsymbol{\theta}) + w_{IC}\mathcal{L}_{IC}(\boldsymbol{\theta}). \qquad (17)$$

Since the scale of the three loss terms can change dramatically during the optimization process, these weights must be dynamic, i.e., their magnitude must be adjusted during the training. In our experiments, we follow the process described in Wang et al. (2021b) for dynamic loss balancing and highlight that this approach is only applicable to cases where Adam is used.

### A 4.3 PHYSICS-INFORMED NEURAL NETWORKS WITH HARD CONSTRAINTS

An alternative approach to dynamic weight balancing is to eliminate $\mathcal{L}_{BC}(\boldsymbol{\theta})$ and $\mathcal{L}_{IC}(\boldsymbol{\theta})$ from Equation (16) by requiring the model's output to satisfy the boundary and initial conditions by construction Berg & Nyström (2018). To this end, we now denote the output of the network by $\widetilde{m}(\mathbf{x}; \boldsymbol{\theta})$ and then formulate the final output of the model as:

$$m(\mathbf{x}; \boldsymbol{\theta}) = a(\mathbf{x})\widetilde{m}(\mathbf{x}; \boldsymbol{\theta}) + b(\mathbf{x}), \qquad (18)$$

where $a(\mathbf{x})$ and $b(\mathbf{x})$ are analytic functions that ensure $m(\mathbf{x}; \boldsymbol{\theta})$ satisfies Equations (15b) and (15c) regardless of what $\widetilde{m}(\mathbf{x}; \boldsymbol{\theta})$ produces at $\mathbf{x}$. A common strategy is to choose $a(\mathbf{x})$ to be the signed distance function that vanishes on the boundaries and produces finite values inside the domain. The construction of $b(\mathbf{x})$ is application-specific since one has to formulate a function that satisfies the applied boundary and initial conditions while generating finite values inside the domain. For the PDE system in Equation (15), one option is $b(\mathbf{x}) = \frac{-2\sin \pi x}{1+e^{-t}}$.

### A 4.4 OPTIMAL RECOVERY

This recent approach leverages zero-mean GPs for solving nonlinear PDEs Chen et al. (2021). Specifically, let us denote the kernel of a zero-mean GP via $c(\cdot, \cdot)$. We associate $c(\cdot, \cdot)$ with the reproducing kernel Hilbert space (RKHS) $\mathcal{U}$ where the RKHS norm is defined as $\|u\|$. Following these definitions, we can approximate $u(\mathbf{x})$ by finding the minimizer of the following optimal recovery problem:

$$\underset{u \in \mathcal{U}}{\text{minimize}} \ \|u\|$$

subject to

$$\begin{aligned}
u_t(\mathbf{x}_i) + u(\mathbf{x}_i)u_x(\mathbf{x}_i) - \nu u_{xx}(\mathbf{x}_i) = 0, & \quad \forall i = 1, \dots, n_{PDE} \qquad (19) \\
u(\mathbf{x}_i) = 0, & \quad \forall i = 1, \dots, n_{BC}, \\
u(\mathbf{x}_i) = -\sin(\pi x_i), & \quad \forall i = 1, \dots, n_{IC},
\end{aligned}$$

where $n_{PDE}$, $n_{BC}$, $n_{IC}$ are the number of nodes inside the domain, on the $x = -1$ and $x = 1$ lines where the boundary conditions are specified, and on the $t = 0$ line where the initial condition is specified, respectively. We denote the collection of these $n_{PDE} + n_{BC} + n_{IC}$ points via $\mathbf{X}$.

The optimization problem in Equation (19) is infinite-dimensional and hence Chen et al. (2021) leverage the representer theorem to convert it into a finite-dimensional one by defining the slack

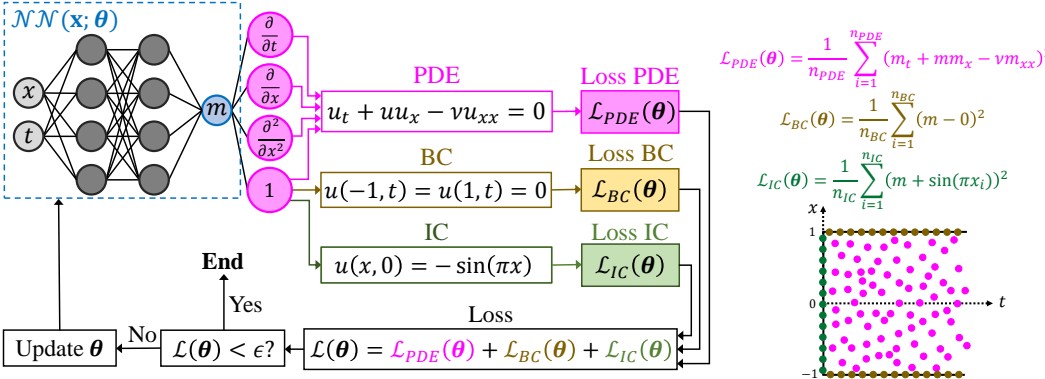

(a) Architecture and loss function for solving the Burgers' equation.

(b) Test points in the domain and on the boundaries.

Figure A2: Physics-informed neural network (PINN): The model parameters, $\boldsymbol{\theta}$, are optimized by minimizing the three-component loss function that encourages the network to satisfy the PDE inside the domain while reproducing the initial and boundary conditions. These loss components are obtained by querying the network on a set of test points that are distributed inside the domain or on its boundaries.

variable $\boldsymbol{z} = \left[ z^{(1)}, z^{(2)}, z^{(3)}, z^{(4)} \right]$:

$$\underset{\boldsymbol{z} \in \mathbb{R}^N}{\text{minimize}} \ \boldsymbol{z}^T \boldsymbol{\Theta}^{-1} \boldsymbol{z}$$

subject to

$$
\begin{aligned}
z_i^{(2)} + z_i^{(1)} z_i^{(3)} - \nu z_i^{(4)} &= 0, & \forall i &= 1, \ldots, n_{PDE} \\
z_i^{(1)} &= 0, & \forall i &= 1, \ldots, n_{BC}, \\
z_i^{(1)} &= -\sin\left(\pi x_i\right), & \forall i &= 1, \ldots, n_{IC},
\end{aligned}
\tag{20}
$$

where $N = 4(n_{PDE} + n_{BC} + n_{IC}) + 3 n_{PDE}$ and $\boldsymbol{\Theta}$ is the covariance matrix (see Section 3.4.1 of Chen et al. (2021) for details on $\boldsymbol{\Theta}$). Equation (20) can be reduced to an unconstrained optimization problem by eliminating the equality constraints following the process described in Subsection 3.3.1 of of Chen et al. (2021). Once $\boldsymbol{z}$ is estimated, the PDE solution can be estimated at the arbitrary point $\mathbf{x}$ in the domain via GP regression.

We note that the process of defining the slack variables and obtaining the equivalent finite-dimensional optimization problem needs to be repeated for different PDE systems (e.g., in a PDE system one may have to define some of the slack variables as the Laplacian of the solution rather than the solution itself). Also, per the recommendations in Chen et al. (2021), $c(\cdot, \cdot)$ is set to an anisotropic kernel and its parameters are chosen manually (i.e., they do not need to be jointly estimated with $\boldsymbol{z}$) but, unlike our approach, this choice must be done carefully since it affects the results. In our comparative studies, we use the values reported in Chen et al. (2021) for the kernel parameters.

A 4.5    IMPLEMENTATION DETAILS IN OUR COMPARATIVE STUDIES

Below, we describe the training procedure of the PIML models used throughout out paper and also comment on how the reference solutions are obtained for each PDE system. All of our codes, data, and models will be made publicly available upon publication.

**Training**    The NN-based approaches (i.e., NN-CoRes, PINN, PINN$_{DW}$, and PINN$_{HC}$) are all implemented in PyTorch Paszke et al. (2019) and use hyperbolic tangent activation functions in all their layers except the output one where a linear activation function is used. The number and size of the hidden layers (see Table 1 in the main text) are exactly the same across these methods to enable a fair and straightforward comparison.

To optimize NN-CoRes, PINNs, and PINN$_{HC}$ we leverage L-BFGS with a learning rate of $10^{-2}$ while PINN$_{DW}$ is optimized using Adam with a learning rate of $10^{-3}$ (note that the performance of L-BFGS deteriorates if dynamic weights are used in the loss function). To ensure these NN-based methods produce optimum models, we use a very large number of epochs during training. Specifically, we employ $1,000$ and $2,000$ epochs for single- and multi-output problems, respectively. Since Adam typically requires more epochs for convergence, we train PINN$_{DW}$ for $40,000$ epochs across all problems. To evaluate the loss function, we use $10,000$ collocation points within the domain in all cases. For PINN and PINN$_{DW}$ we uniformly sample boundary and/or initial conditions at $1,000$ locations while we only sample $40$ points for NN-CoRes. This significant difference is due to the fact that we observed that NN-CoRes with just $40$ boundary points can outperform other methods. Leveraging more boundary data improves the performance of NN-CoRes in solving PDE systems especially in satisfying the boundary and initial conditions.

We fit GP$_{OR}$ based on the code and specifications provided by Chen et al. (2021) which leverages a variant of the Gauss–Newton algorithm for optimization. The performance of GP$_{OR}$ depends on the kernel parameters and the number of interior nodes $n_{PDE}$ where $\boldsymbol{z}$ needs to be estimated. For the former, we use the recommended values in Chen et al. (2021) and for the latter we choose two values ($1,000$ and $2,000$) in our experiments.

NN-CoRes, PINN, PINN$_{DW}$, and PINN$_{HC}$ are trained on an NVIDIA GeForce RTX 3060 with 64 GB of RAM whereas GP$_{OR}$ is trained on a CPU equipped with a 11th Gen Intel-Core i7-11700K running at a base clock speed of 3.6 GHz.

**Reference Solutions**    We obtain the reference solutions for the PDE systems as follows:

Table A1: Summary of comparative studies for the LDC problem: We report $L_e^2 \times 10^3$ of different methods as a function of model capacity and $A$. The symbol $\otimes$ indicates the network architecture (e.g., $4 \otimes 10$ is an NN which has four $10-$ neuron hidden layers). Unlike NN-based methods, $\text{GP}_{\text{OR}}$'s accuracy relies on the number of interior nodes which we set to $1,000$ or $2,000$. $\text{GP}_{\text{OR}}$ is not applied to LDC as it relies on manual derivation of the equivalent variational problem which, unlike the first three PDEs, is not done by the developers Chen et al. (2021).

| | | NN-CoRes | | $\text{GP}_{\text{OR}}$ | | PINN | | $\text{PINN}_{\text{DL}}$ | | $\text{PINN}_{\text{HC}}$ | |
|---|---|---|---|---|---|---|---|---|---|---|---|
| Problem | Capacity | $4 \otimes 10$ | $4 \otimes 20$ | 1,000 | 2,000 | $4 \otimes 10$ | $4 \otimes 20$ | $4 \otimes 10$ | $4 \otimes 20$ | $4 \otimes 10$ | $4 \otimes 20$ |
| | $u$ | 192 | **85.6** | – | – | 266 | 123 | 299 | 126 | 399 | 497 |
| LDC ($A = 3$) | $v$ | 174 | **82.5** | – | – | 278 | 128 | 307 | 123 | 306 | 395 |
| | $p$ | 191 | **91.9** | – | – | 272 | 133 | 298 | 125 | 592 | 696 |
| | $u$ | 249 | **222** | – | – | 601 | 567 | 597 | 518 | 1019 | 757 |
| LDC ($A = 5$) | $v$ | 251 | **220** | – | – | 632 | 592 | 629 | 543 | 704 | 564 |
| | $p$ | 433 | **394** | – | – | 917 | 872 | 923 | 809 | 1362 | 1411 |

- Burgers' Equation: The reference solution is obtained from the code provided in Chen et al. (2021) which employs the Cole-Hopf transformation Ohwada (2009) together with the numerical quadrature.

- Elliptic PDE: The analytical solution for this problem is $u(x, y) = \sin(\pi x)\sin(\pi y) + 2\sin(4\pi x)\sin(\pi y)$.

- Eikonal Equation: We leverage the solution method provided by Chen et al. (2021) which applies the transformation $u(x, y) = -\epsilon \log g(x, y)$ leading to the linear PDE $g - \epsilon^2 \Delta g = 0$ that can be solved via the finite difference method.

- Lid-Driven Cavity: we use the finite element method implemented in the commercial software package COMSOL Multiphysics (1998).

## A 5 ADDITIONAL EXPERIMENTS

In the following subsections, we summarize the findings of some additional experiments that emphasize the attractive properties of NN-CoRes. We first provide some details on prediction errors that complement the results reported in the main text. Afterwards, we elaborate on the extended version of our framework that accommodates PDE systems such as the Navier-Stokes equations that have multi-variate solutions. Then, we conduct rigorous sensitivity analyses to characterize the effect of factors such as random initialization, noise (on data obtained from the initial and/or boundary conditions), optimization settings, and architecture on our results. These analyses demonstrate that our framework is substantially less sensitive to such factors compared to competing methods. We conclude this section by investigating the behavior of NN-CoRes' loss function during training and extending our approach for solving inverse problems.

### A 5.1 DETAILED ERROR ANALYSIS

The solution of the LDC problem consists of three dependent variables which are the pressure $p(\mathbf{x})$ and the two velocity components in the $x$ and $y$ directions, $u(\mathbf{x})$ and $v(\mathbf{x})$, respectively. In Table 1 we report the mean of the Euclidean norm of the error on the three outputs. In Table A1 we provide the errors for the individual outputs of this benchmark problem and observe the same trend where NN-CoRes consistently outperforms other methods. We also notice that all the models predict pressure with less accuracy compared to the velocity components. This trend is due to the facts that not only the scale of $p(x, y)$ is smaller than the velocity components, but also $p(x, y)$ is known at a single point on the boundaries whereas $u(x, y)$ and $v(x, y)$ are known everywhere on the boundaries.

To gain more insight into the performance of each method, we visualize the error maps in Figure A3. We observe that $\text{GP}_{\text{OR}}$ is least accurate either in regions with sharp solution gradients or inside the domain where boundary information is not effectively propagated inward by the zero-mean GP. For $\text{PINN}_{\text{DW}}$, the errors are predominantly close to either the boundaries or where solution discontinuities are expected to appear. PINNs' errors in reproducing the BCs/IC is eliminated in $\text{PINN}_{\text{HC}}$ but at the expense of significant loss of accuracy elsewhere in the domain. These issues are

largely addressed by NN-CoRes which reproduce BCs/IC and approximate high gradient solutions quite well.

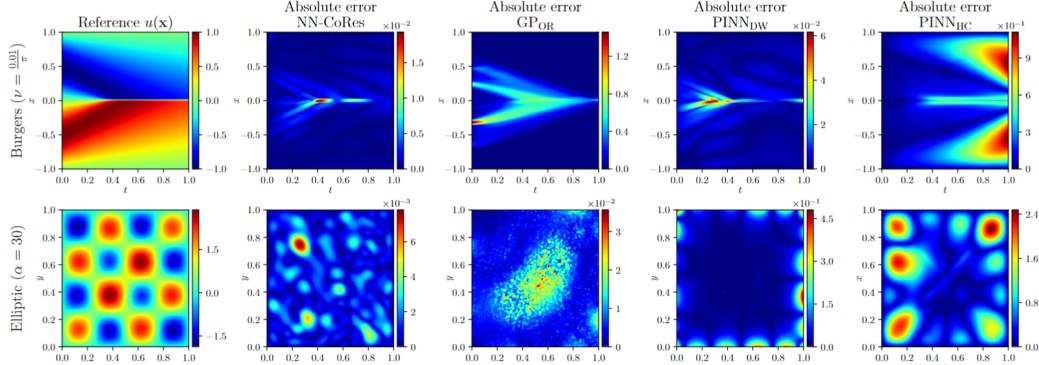

Figure A3: Reference solutions and absolute error maps: Error maps of NN-CoRes are consistently smaller than the other three methods.

Similar to Figure A3, we provide the reference solution and error maps of different approaches for the Eikonal problem in Figure A4a where we observe similar patterns. Specifically, $GP_{OR}$ fails to properly propagate the boundary information inwards as it relies on a zero-mean GP. PINN is quite accurate inside the domain but cannot faithfully satisfy the boundary conditions. $PINN_{DW}$ addresses the inaccuracy of PINN close to the boundaries but incurs significant errors inside the domain as the reformulation in Equation (18) complicates the training dynamics. These issues are effectively addressed by NN-CoRes which achieve small errors inside the domain and on the boundaries.

In Figure A4b we solve a canonical PDE system known as Helmholtz Wang et al. (2021b) which is defined as:

$$
\begin{aligned}
u_{xx}(x,y) + u_{yy}(x,y) + u(x,y) &= q(x,y), & \forall x,y \in (-1,1)^2 \\
u(x,-1) = u(x,1) &= 0, & \forall x \in [-1,1] \\
u(-1,y) = u(1,y) &= 0, & \forall y \in [-1,1]
\end{aligned}
\tag{21}
$$

In Equation (21), $q(x,y)$ is constructed such that the analytic solution is $u(x,y) = \sin(a_1\pi x)\sin(a_2\pi y)$ where $a_1$ and $a_2$ are two constants that control the frequency along the $x$ and $y$ directions, respectively. The Helmholtz equation is a well-studied benchmark problem since PINNs fail to accurately solve it. To address this shortcoming, recent works have introduced quite complex architectures which typically leverage adaptive loss functions. We test our framework on this benchmark problem by setting $a_1 = 1$ and $a_2 = 4$ while using the same architecture and training procedure that are used in our comparative studies. As shown in Figure A4b our predictions accurately capture both the high- and low-frequency features of the solution. We note that the solution in Figure A4b is 5 times more accurate than the one reported in Wang et al. (2021b) which employs a considerably larger architecture $(4 \otimes 50)$ and leverages the adaptive loss function described in Equation (17).

## A 5.2    EXTENSION TO COUPLED SYSTEMS

As schematically illustrated in Figure A5 we slightly modify our framework to solve coupled PDE systems such as the Navier-Stokes equations which have multiple dependent variables that interact with one another. The essential idea behind this modification is to endow each dependent variable with a GP prior. These GPs have independent kernels but a shared mean function that is parameterized via a deep neural network. While a single kernel can help in learning the inter-variable relations, we avoid this formulation for two main reasons. Firstly, it increases the size and condition number of the covariance matrix especially if the boundary conditions on these variables are significantly different. For instance, on the top edge $(y = 1)$ in the LDC benchmark problem, pressure is unknown while the vertical and horizontal velocity components are equal to, respectively, zero

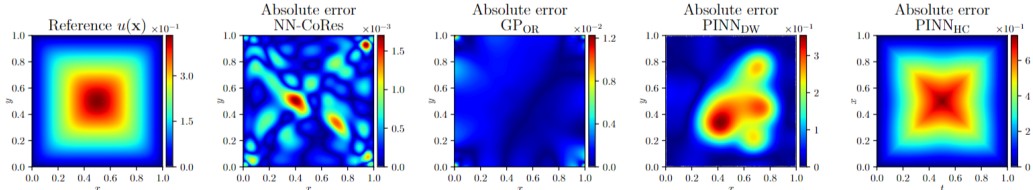

(a) Reference solution and error maps of different approaches for the Eikonal equation with $\epsilon = 0.05$ with a $4 \otimes 20$ architecture.

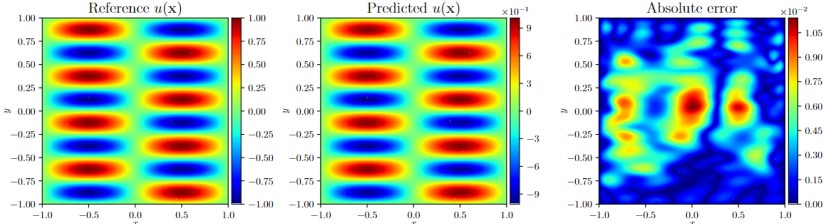

(b) Solving the high-frequency Helmholtz equation via NN-CoReswith a $4 \otimes 20$ architecture.

Figure A4: Reference solutions and error maps: Our approach provides much lower errors compared to other methods and automatically adapts to high- and low-frequency solutions.

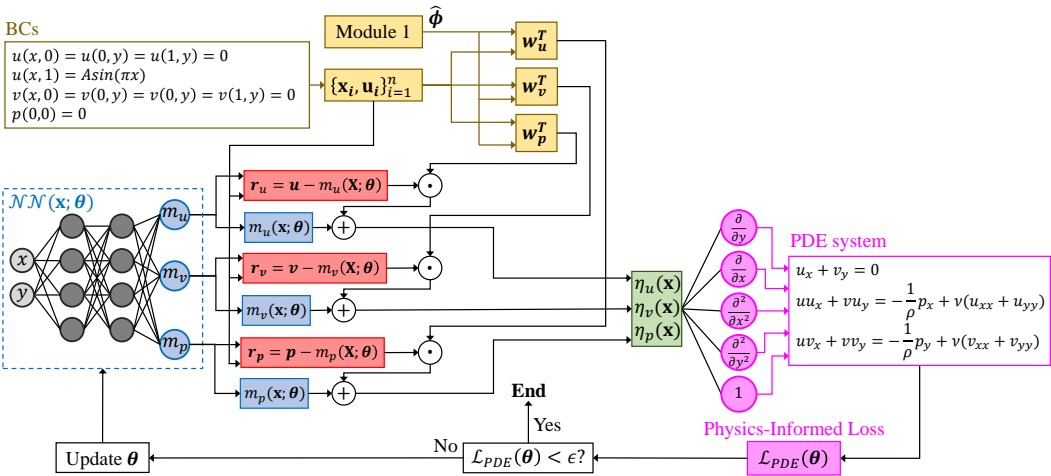

Figure A5: Solving the 2D incompressible Navier-Stokes equations for the lid-driven cavity problem: With minor architectural changes, our framework can also solve coupled PDE systems. Specifically, we endow each dependent variable with a GP prior. These GPs have independent kernels but a shared mean function that is parameterized via a deep neural network.

and $A \sin(\pi x)$. Secondly, our empirical findings indicate that the shared mean function is able to adequately learn the hidden interactions between these dependent variables.

## A 5.3 SENSITIVITY ANALYSES

In this section, we conduct a wide range of sensitivity studies to assess the impact of factors such as random initialization, noise, network architecture, and optimization settings on the summary results reported in the main text.

We first analyze the effect of the roughness parameter, $\omega$, on the results. We use the simple Gaussian kernel in Equation (3) with $\sigma^2 = 1$ and $\omega = 2$ in all of our studies. The nugget or jitter parameter of the kernel is chosen such that the covariance matrix is numerically stable. We ensure this stability by

imposing an upper bound of approximately $\kappa_{max} \approx 10^6$ on the condition number of the covariance matrix, i.e., $\kappa < \kappa_{max}$. This constraints typically results in a nugget value of around $10^{-5}$ or $10^{-4}$. We have not optimized the performance of NN-CoRes with respect to $\kappa_{max}$ as we have found our current results to be sufficiently accurate.

As stated in the main text, the performance of NN-CoRes is quite robust to the values chosen for $\phi = 10^\omega$ as long as they lie within a certain range. To obtain this range, we conduct the following inexpensive experiment using the Burgers' equation and the extension of the kernel in Equation (3) to two-dimensional inputs, i.e., $c(\mathbf{x}, \mathbf{x}'; \phi, \delta, \sigma^2 = 1) = \exp\{-\phi(x - x')^2 - \phi(t - t')^2\} + \mathbb{1}\{x == x'\}\delta$. We first sample $n_{train}$ equally spaced boundary samples using the provided analytic initial and boundary conditions. To quantify the effect of data size on the results, we consider 5 scenarios where $n_{train} \in \{10, 20, 40, 80, 160\}$. For each of these five cases, we build 200 independent GPs whose only difference is the value that we assign to $\omega$. Specifically, we consider 200 equally spaced values in the $[-2, 6]$ range for $\omega$ and use each of these values in one of the GPs which all have a non-zero mean function (we use a deep NN whose parameters are randomly initialized and frozen as the mean function). Once these GPs are built, we use them to predict on $n_{test} = 10^4$ boundary points (see Equation 1 in the main text for the prediction formula). The results of this study are shown in the left and middle plots in Figure A6a and indicate that as more training data are sampled on the boundaries a wider range of values for $\omega$ result in small test errors. We highlight that this study is computationally very fast since none of the GPs are optimized; rather their parameters are either chosen by us (i.e., $\omega$), or fixed (i.e., $\delta, \sigma^2$, and parameters of the NN mean).

Following the above study, we have decided to use 40 boundary points in NN-CoRes. Based on the left and middle plots in Figure A6a, $\omega = 2$ seems to be a good choice (but not the optimum one) for minimizing the error in reproducing the initial and boundary conditions. To see the effect of this choice on the performance of a trained NN-CoRes, we again vary $\omega$ (50 equally spaced values in the $[-2, 6]$ range) but this time we train an NN-CoRes model for each value of $\omega$. We evaluate the performance of these models in solving the Burgers' equation by reporting the Euclidean norm of the error $L_2^e$ at $n_{test} = 10^4$ points randomly located in the domain. The results are shown in the right plot in Figure A6a and indicate that although $\omega = 2$ is not the optimum choice, it yields a model whose performance is close to optimal (the optimum model is achieved via an $\omega$ close to 3).

We now conduct a few extensive experiments to study the effect of network size and optimization settings on the performance of various NN-based models. First, we fix everything and increase the number of neurons in each hidden layer from 10 to 50 (at increments of 10) and solve the Burgers' and Elliptic PDEs via both NN-CoRes and PINNs. We then repeat this experiment but this time we fix the architecture to $4 \otimes 20$ and incrementally increase $n_{PDE}$ from $10^3$ to $10^4$. The results of these two experiments are summarized in Figure A6b and indicate that NN-CoRes is much less sensitive to the problem than PINNs which perform quite well on Burgers' but fail at accurately solving the Elliptic PDE that has direction-dependent frequency. We also observe that NN-CoRes provide lower errors than PINNs in most simulations.

In our next experiment, we study the effects of optimizer (L-BFGS vs Adam), random initialization, and architecture type on the performance of various models. To this end, we again consider the Burgers' and Elliptic PDE systems and solve them with six NN-based methods and GP$_{OR}$. For each case we repeat the training process of each model 10 times to quantify the effect of random initialization on the models' solution accuracy. For these experiments, we also consider a new network architecture that we denote by M3 which is introduced in Wang et al. (2021b) and aims to improve gradient flows by designing feed-forward networks with connections that resemble transformers Vaswani et al. (2017). In our framework, we replace the architecture that is used in all of our studies (which is a feed-forward neural network or an FFNN) with M3 and train the model with Adam (the resulting model is denoted by M3-CoRes). We also train another NN-based model denoted by M4 Wang et al. (2021b) whose architecture is the same as M3 but leverages dynamic weights in its loss function. We highlight that the simulations that leverage M3 as their architecture have more parameters (and hence learning capacity) than cases where FFNNs are used so we expect M3-based simulations to provide lower errors.

The results of these simulations are summarized in Figure A6c and indicate that (1) NN-CoRes and GP$_{OR}$ are less sensitive to random initializations compared to PINNs and their variations, (2) unlike other models, NN-CoRes performs well in both PDE systems, i.e., our framework provides a more

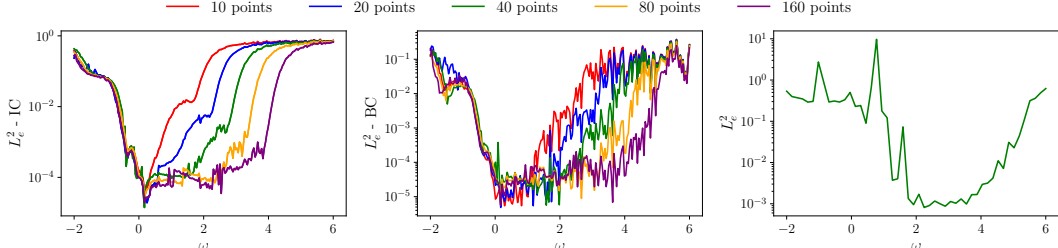

(a) Effect of $\omega$ on GP's interpolation power (left and middle plots) and NN-CoRes (right plot). Burgers' equation is used in this study.

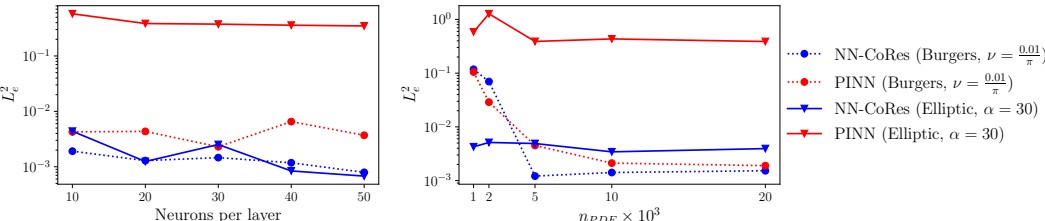

(b) Effect of network size (left, $n_{PDE} = 10^4$) and $n_{PDE}$ (right, $4 \otimes 20$ architecture) on the accuracy of PINNs and NN-CoRes.

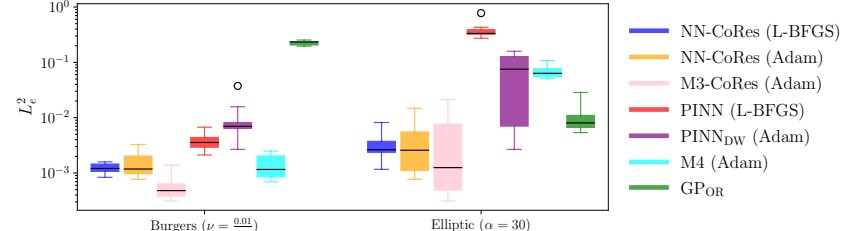

(c) Effect of optimizer, random initialization, and architecture type on errors for the Burgers' and Elliptic problems.

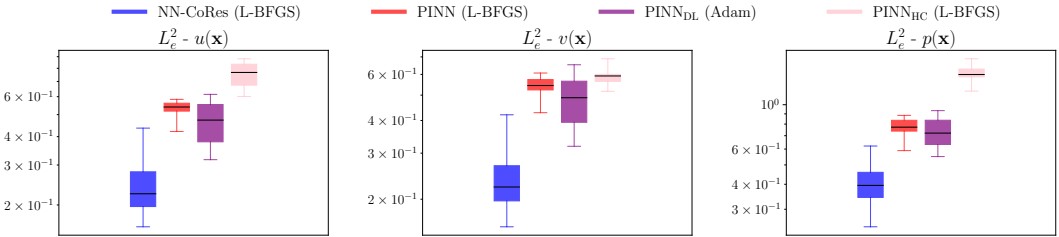

(d) Effect of random initialization and optimizer on errors for the LDC problem ($A = 5$). All models have $4 \otimes 20$ architecture and use $n_{PDE} = 10^4$.

Figure A6: Sensitivity studies: We analyze the sensitivity of our results to factors such as the roughness parameters in the kernel, optimization settings, network architecture, and initialization. Based on these experiments, NN-CoRes provide a more robust machine learning-based approach for solving different nonlinear PDEs.

transferable method for solving PDEs via machine learning, and (3) architectures besides simple FFNNs (such as M3) can also be used in our framework to achieve higher accuracy.

The above experiments are based on the Burgers' and Elliptic PDE problems but our studies indicate that similar trends appear in other problems. To demonstrate this, we solve the LDC problem via four NN-based models that either use L-BFGS or Adam as their optimizer. We repeat the training

Figure A7: Reference vs predictions (LDC with $A = 5$): Performance improves as the network sizes increase. The small NN-CoRes is more accurate than the large PINN$_{\text{DW}}$.

process of each model 10 times to assess the effect of random parameter initialization on each model's performance. The results are summarized with the boxplots in Figure A6d and agree with our previous findings that indicate NN-CoRes consistently outperform other methods.

Comparing Figure A6c and Figure A6d we observe that the errors reported for the LDC problem are consistently larger than those reported for the Burgers' and Elliptic problems. The reason behind this trend is that not only LDC is a more complex problem where the PDE solution consists of three inter-dependent variables (compared to only one variable in the case of Burgers' or Elliptic PDEs), but also pressure is only known at a single point on the boundary (rather than everywhere on the boundary). To test the first assertion, we increase the networks sizes for both NN-CoRes and PINN$_{\text{DW}}$ and observe in Figure A7 that both models provide higher accuracy compared to the reported numbers in Table 1. We highlight that the percentage of improvement is noticeably higher in the case of NN-CoRes and these errors keep reducing as the network size increases, as demonstrated in Figure A8.

Finally, we investigate the effect of noisy boundary data on our results. Specifically, we corrupt the solution values that we sample from the initial and/or boundary conditions before using them in our approach. We use a zero-mean normal distribution to model the noise and set the standard deviation to either $0.5\%$ or $1\%$ of the solution range. As shown in Figure A9, the solution accuracy decreases as the noise variance increases (this trend is expected) but in all cases NN-CoRes are able to quite effectively eliminate the noise and solve the Burgers' and Elliptic PDE systems.

## A 5.4 LOSS AND ERROR BEHAVIOR

To gain more insights into the training dynamics of our approach, we visualize the loss and accuracy during the training process in Figure A10 and Figure A11 where in the latter figure we track the errors individually for each output. We provide these plots for both PINNs and NN-CoRes where the loss function of the former is based on Equation (16) while NN-CoRes only use $\mathcal{L}_{PDE}(\boldsymbol{\theta})$ in their loss function. The solution accuracy is measured based on $L_e^2$ and $(L_e^2)^2$ for points inside the domain and on its boundaries. Note that we square $L_e^2$ on the boundaries to be able to directly see its contribution to PINNs' loss, see $\mathcal{L}_{BC}(\boldsymbol{\theta})$ in Equation (16). In the case of NN-CoRes, we also report the accuracy of its NN part on predicting the PDE solution to quantify the contributions of kernel-weighted CoRes towards the model's predictions.

As it can be observed in Figures A10 and A11, NN-CoRes typically converge faster than PINNs, see the plots whose $y-$axis title is $L_e^2$ - Domain. We attribute this trend to the fact that, unlike in PINNs, the initial and boundary conditions are automatically satisfied in our models thanks to the kernel-weighted CoRes which are smooth functions. This features enables NN-CoRes to focus on satisfying the PDE system in module two of our framework while PINNs have to struggle with both the differential equations as well as the initial and boundary conditions.

An interesting trend in Figures A10 and A11 is that the errors of NN-CoRes are consistently lower than their NN components both in the domain and on the boundaries. That is, the kernel-weighted CoRes positively contribute to the model's predictions both on the boundaries and inside the domain. This behavior is in sharp contrast to most approaches such as PINN$_{\text{HC}}$ that satisfy the boundary conditions at the expense of complicating the training process.

Another interesting trend that we observe in Figures A10 and A11 is that PINNs achieve lower loss values than NN-CoRes in the case of Eikonal and LDC problems. While lower loss values are desirable, in these cases the observed trends are misleading. To explain this behavior, we note that the

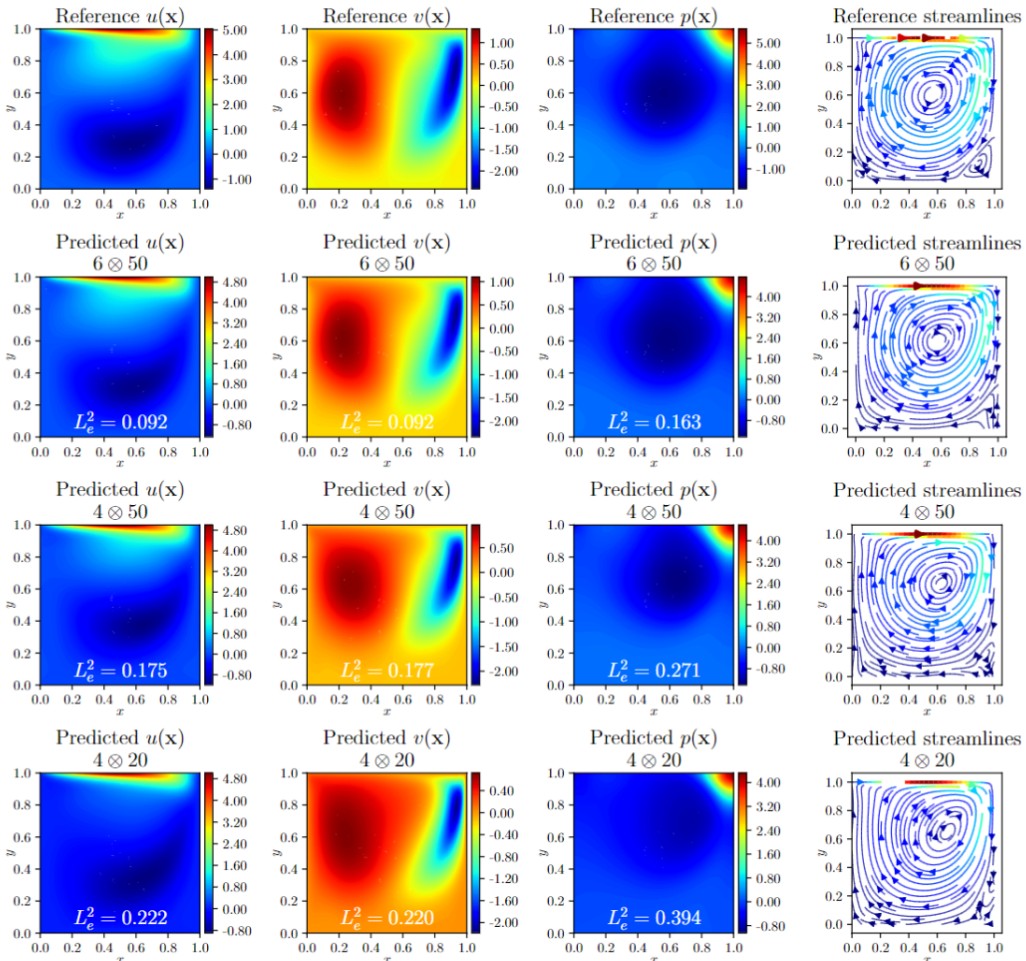

Figure A8: Effect of network size on the accuracy in the LDC problem: The accuracy of NN-CoRes consistently increases in predicting $u, v$, and $p$ as the network size (either the number of hidden layers or their size) increases.

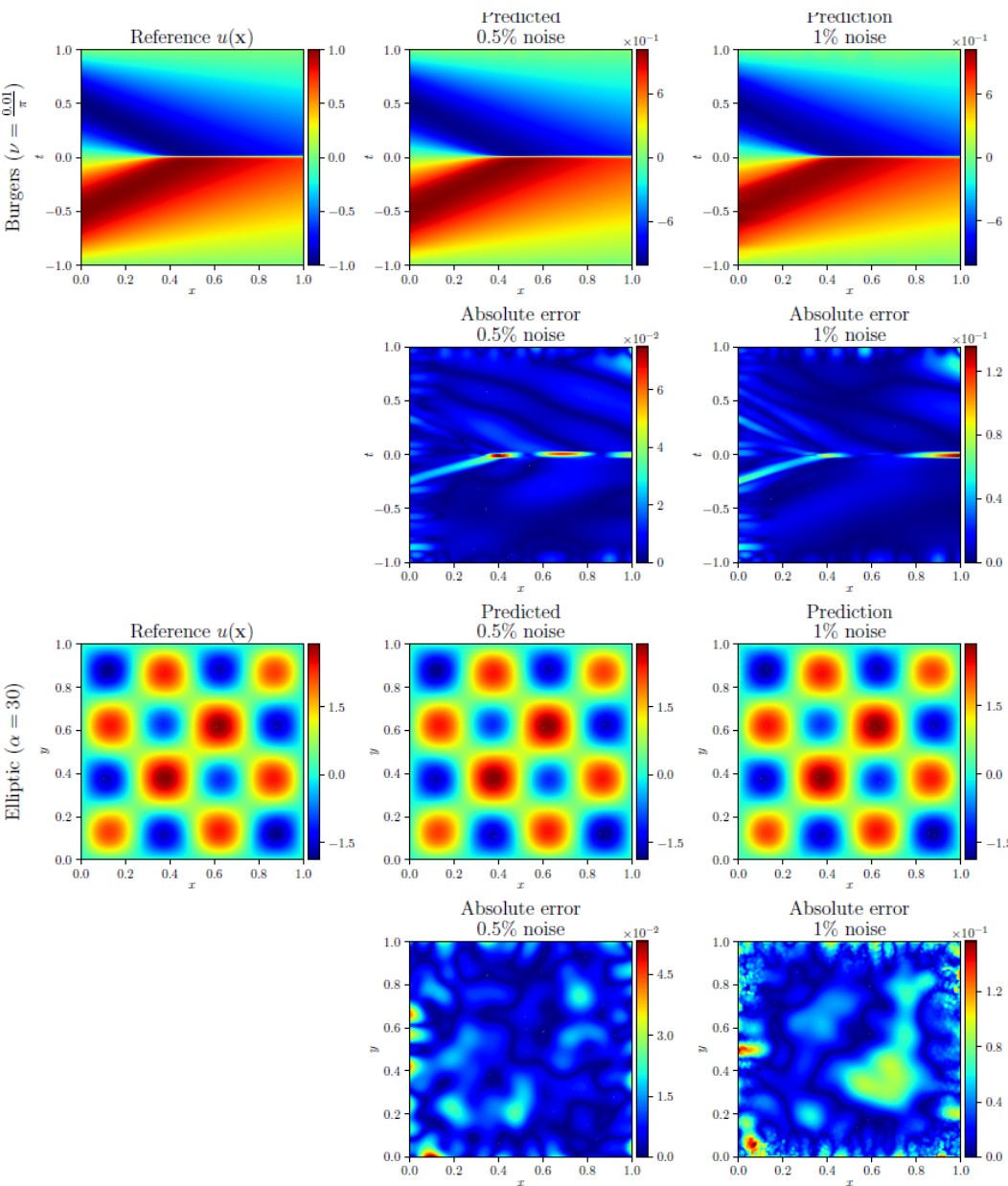

Figure A9: Reference and predicted solutions with noisy boundary data: We corrupt the samples obtained from boundary and initial conditions by either $0.5\%$ or $1\%$ of the solution range. In all cases, the performance of NN-CoRes is insignificantly affected by the noise.
.

loss function of NN-CoRes is simply $\mathcal{L}_{PDE}(\boldsymbol{\theta})$ as the boundary and initial conditions are automatically satisfied. However, the loss function of PINNs minimizes both $\mathcal{L}_{PDE}(\boldsymbol{\theta})$ and $\mathcal{L}_{BC}(\boldsymbol{\theta})$. That is, since PINNs do not strictly satisfy the boundary conditions, they are less regularized and hence can minimize $\mathcal{L}_{PDE}(\boldsymbol{\theta})$ (which dominates the overall loss) in a more flexible manner. However, this behavior provides less accuracy since the boundary conditions are not learnt sufficiently well.

Finally, to visually compare the capacity utilization across different NN-based models, in Figure A12 we provide the histogram of the PDE loss gradients with respect to $\boldsymbol{\theta}$ at the end of training. We observe that NN-CoRes achieve the most near-zero gradients *while* satisfying the BCs/IC. In contrast, PINN$_{HC}$, which is also designed to automatically satisfy the BCs/IC, struggles to minimize the PDE loss.

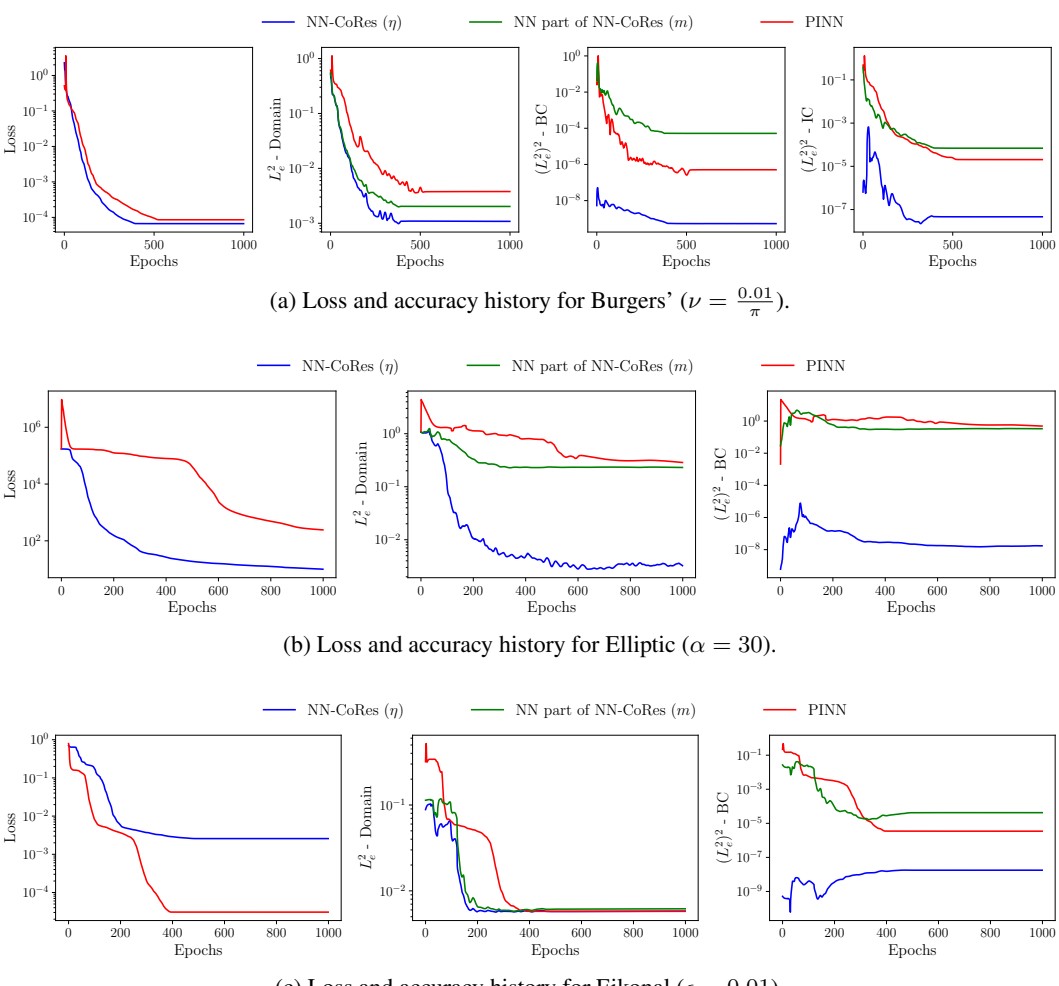

(a) Loss and accuracy history for Burgers' ($\nu = \frac{0.01}{\pi}$).

(b) Loss and accuracy history for Elliptic ($\alpha = 30$).

(c) Loss and accuracy history for Eikonal ($\epsilon = 0.01$).

Figure A10: Loss convergence and error decomposition: NN-CoRes typically converge faster than PINNs and consistently provide more accurate solutions. The NN part of NN-CoRes benefits from the kernel-weighed CoRes not only on the boundaries, but also inside the domain.

## A 5.5 INVERSE PROBLEMS

So far we have only used the differential equations along with the initial and boundary conditions in building NN-CoRes. In this section, we introduce an extension of our framework for solving inverse problems where (1) there are some (possibly noisy) labeled data available inside the domain (we refer to these samples as observations to distinguish them from data obtained from the initial and/or boundary conditions), and (2) one or more parameters in the differential equations are unknown. Our goal in such applications is to solve the PDE system while estimating the unknown parameters.

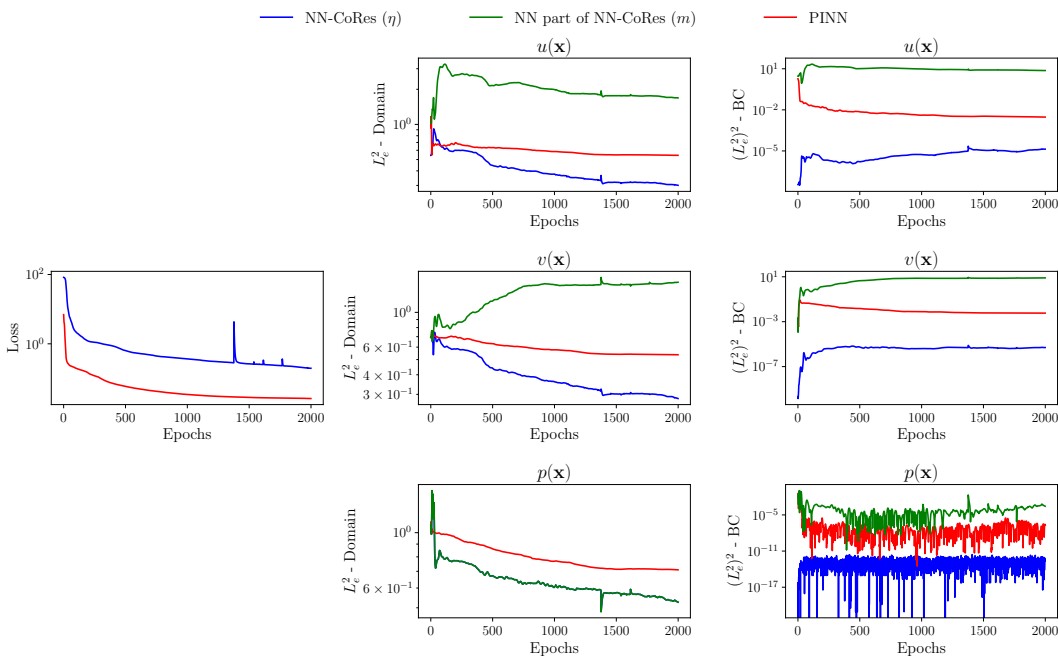

Figure A11: Loss convergence and error decomposition for LDC: The NN part of NN-CoRes benefits from the kernel-weighed CoRes not only on the boundaries, but also inside the domain. In the case of pressure, kernel-weighed CoRes do not contribute to the model's predictions as $p(\mathbf{x})$ is only known at a single point on the boundary.

As shown in Figure A13a, we modify our framework in two ways to solve the PDE system in Equation (15) assuming $\nu$ is unknown but $u(\mathbf{x})$ is known at $n_{obs}$ random points in the domain. Specifically, we (1) use the $n_{obs}$ observations in the kernel of NN-CoRes in exactly the same way that the $n_{BC} + n_{IC}$ boundary data are handled by the kernel, and (2) treat $\nu$ as one additional parameter that must be optimized along with the weights and biases of the NN.

To evaluate the performance of our approach in solving inverse problems, we consider the Burgers', Elliptic, and Eikonal PDE systems. We solve each problem in two scenarios where there are either $n_{obs} = 100$ or $n_{obs} = 200$ observations available in the domain. As shown in Figure A13b, in all cases NN-CoRes can estimate the unknown PDE parameter quite accurately. The convergence rate in all cases is quite fast and insignificantly reduces as $n_{obs}$ is halved from 200 to 100.

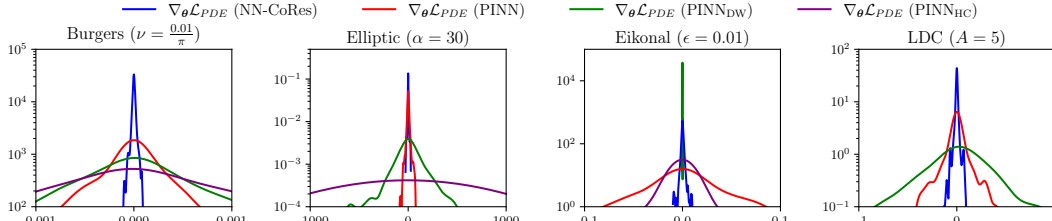

Figure A12: Histograms of PDE loss gradients: NN-CoRes is more effective in satisfying the PDE system. While PINN$_{DW}$ has more near-zero gradients in the Eikonal problem, it does so at the expense of violating the BC loss term, which leads to a more inaccurate solution as it is reflected in Figure A10. All models in this figure have a $4 \otimes 20$ architecture.

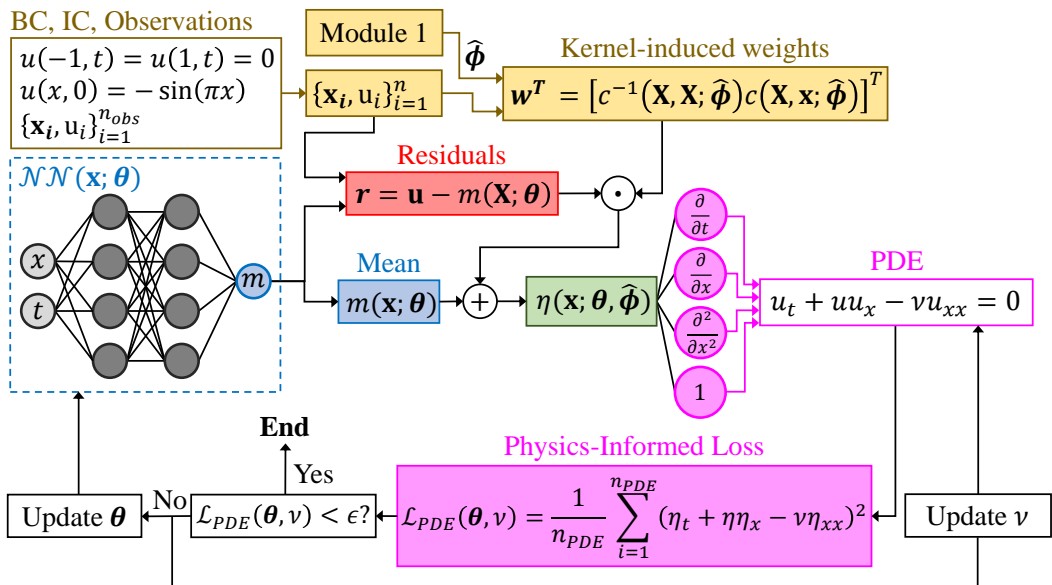

(a) Flowchart of module two of our framework for solving inverse problems. The flowchart is tailored to the PDE system in Equation (15).

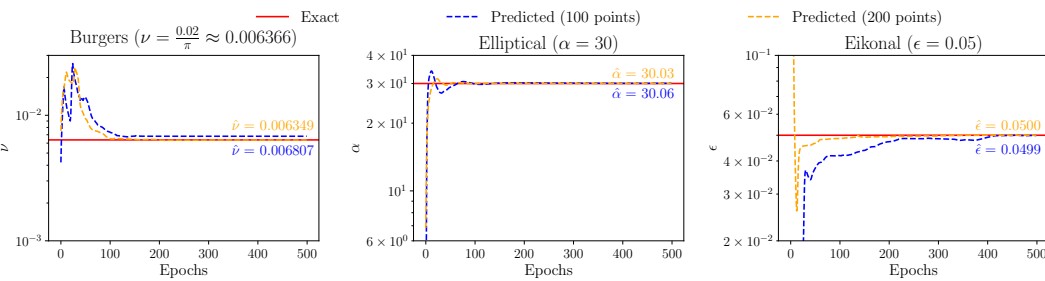

(b) Convergence rates are fast and improve as more data are infused into our model.

Figure A13: Inverse problems via NN-CoRes: We modify the flowchart in Figure 1 in two ways to solve a PDE system whose one or more parameters may be unknown. NN-CoRes treat observations (i.e., labeled solution data inside the domain) identically to boundary data and are very effective in using them in estimating the unknown PDE parameters.

