# OpenReview forum: "Integrating Kernel Methods and Deep Neural Networks for Solving PDEs"
_ICLR.cc/2024/Workshop/AI4DiffEqtnsInSci — AI4DiffEqtnsInSci @ ICLR 2024 Poster_

### Official Review · Reviewer_8ReM · 2024-03-03
**Nice paper and contribution**

**Rating:** 7
**Confidence:** 4

**Review:**

## Summary
This paper proposes to merge kernel methods with DNNs with applications to solving PDEs. In particular, they propose Corrective Residuals (CoRes).

## Strengths
- Nice theoretical analysis and guarantees of the proposed method
- Clear overview of SciML methods in the intro and references
- Nice experimental results on broad set of canonical PDEs including the hyperbolic Burgers and Eikonal equation.
- The paper is well-written.
- Very important that the proposed methods satisfies the BCs/IC as the number of sampled
boundary points increases and is independent of the domain geometry and the potential noise corrupting the data. Typically this is ignored in many ML approaches, so it is nice to see the authors include it with theoretical proofs.
- Very detailed appendix

## Weaknesses
- Please provide reference to the first sentence of sentence 3 that kernel methods such as GPs have less extrapolation and scalability powers compared to deep NNs.
- Baselines: I think kernel based Neural Operator methods, e.g., FNO (Li et al., 2020) and Multi-wavelet Neural Operator (Gupta et al., 2021) should be compared to as well since these methods also use a kernel approach within DNNs rather than just comparing to GPs and PINNs

---

### Official Review · Reviewer_p7T5 · 2024-03-03
**An okay paper with critical experiments missing**

**Rating:** 5
**Confidence:** 4

**Review:**

This work talks about using kernels in physics-informed ML setting. The problem setup is interesting as it incorporates IC and BC together with the PDE. However, the reviewer has the following issues:
1. Lack of proper evaluation. It seems like only 0 Dirichlet BC is studied in the paper. The other cases such as non-zero Dirichlet, periodic, Neumann etc. should also be addressed, or at least mentioned.
2. It is not clear why robustness to optimizer is valid here. An intuitive reasoning would strengthen the work. Apart from Fig-A6c, there is no other mention that reviewer could find.

---

### Meta-Review · Area_Chair_H3ei · 2024-03-03

**Recommendation:** Accept (Poster)

**Metareview:**

The reviewer mentions that this paper proposes an interesting approach to combine classical kernel methods with DNNs. In particular, the proposed method improves the performance of GPs and PINNs. Another advantage over classical PINNs that the authors are able to satisfy the initial and boundary conditions exactly rather than as soft constraints in the loss function. I think the comments from Reviewers 8ReM and p7T5 should be addressed in the camera-ready version and I vote for acceptance.

---

### Decision · Program_Chairs · 2024-03-03

Accept (Poster)